



# Measurements and modelling of airborne plutonium in Subarctic Finland
# between 1965 and 2011
Susanna Salminen-Paatero*[a,b], Julius Vira [b], Jussi Paatero [b]
a) Department of Chemistry, Radiochemistry, P.O. Box 55, FI-00014 University of Helsinki, Finland
(Present address). susanna.salminen-paatero@helsinki.fi.
b) Finnish Meteorological Institute, P.O. Box 503, FI-00101 Helsinki, Finland. julius.vira@fmi.fi,
jussi.paatero@fmi.fi.
* Corresponding author.
ABSTRACT
The activity concentrations of $^{238,239,240}$Pu and $^{241}$Am (for determining its mother nuclide $^{241}$Pu) as
well as activity ratios $^{238}$Pu/$^{239+240}$Pu, $^{241}$Pu/$^{239+240}$Pu, $^{239+240}$Pu/$^{137}$Cs, and mass ratio $^{240}$Pu/$^{239}$Pu were
determined from air filter samples collected in Rovaniemi (Finnish Lapland) in 1965-2011. The origin
of plutonium in surface air was assessed based on this data from long time series. The most important
Pu sources in the surface air of Rovaniemi were atmospheric nuclear weapons testing in 1950's and
1960's, later nuclear tests in 1973-1980, and SNAP-9A satellite accident in 1964, whereas the
influence from the 1986 Chernobyl accident was only minor. Contrary to the alpha emitting Pu
isotopes, $^{241}$Pu from the Fukushima accident in 2011 was detected in Rovaniemi. Dispersion
modelling results with the Silam model indicate that Pu contamination in northern Finland due to
hypothetical reactor accidents would be negligible in case of a floating reactor at the Shtokmann
natural gas field and relatively low in case of an intended nuclear power plant in western Finland.
Key words: Plutonium, isotope ratio, Chernobyl, nuclear weapons testing, nuclear accident,
dispersion modelling





1. INTRODUCTION

Distribution of anthropogenic radionuclides in global fallout from nuclear weapons testing is uneven,
and even more inhomogeneous is their distribution in regional and local fallout from different sources.
It is known that Subarctic and Arctic regions have received radionuclide deposition with different
radioactivity level and composition than the more temperate areas of the Earth. Subarctic and Arctic
ecosystems have a special combination of harsh climate, often sparse vegetation, lack of nutrients
and, in case of humans, dependence on traditional lines of business and life styles like hunting,
fishing, reindeer herding, and collection of mushroom and berries. Consequently, these Nordic
ecosystems are highly vulnerable for toxic agents, including radionuclides. Still there are only few
contiguous long-term radioactivity data series from Subarctic and Arctic areas where the changes in
concentration levels and isotope ratios could be followed and different nuclear events identified as
contamination sources in a particular environment.
In total, radionuclides $^{137}$Cs, $^{90}$Sr, total beta activity, $^{238,239,240}$Pu and $^{241}$Am were determined from the
air filter samples that were collected in Rovaniemi (Finnish Lapland) in 1965-2011. $^{241}$Am (t½ 432.2
a) was analyzed for calculating the activity concentration of its mother nuclide, relatively short-lived
beta emitter $^{241}$Pu (t½ 14.35 a). The major part of $^{241}$Am in the samples originates from the decay of
$^{241}$Pu after the sampling and only a minor part of $^{241}$Am originates directly from nuclear events. The
results for $^{137}$Cs, $^{90}$Sr, and total beta activity have been reported elsewhere (Salminen-Paatero et al.
2019). The activity ratio $^{238}$Pu/$^{239+240}$Pu and the mass ratio $^{240}$Pu/$^{239}$Pu in Rovaniemi have been
presented pictorially with other global ratio values in the article by Thakur et al. (2017), but the ratio
values of Rovaniemi were not discussed in detail there.
In this work, radionuclide concentration and isotope ratio data from 1965-2011 has been used for
estimating nuclear contamination sources in the surface air of Finnish Subarctic during almost five
decades. Furthermore, atmospheric dispersion of one real and one hypothetical nuclear events has
been modelled for finding out potential transport of Pu isotopes and effect of these nuclear events on
atmospheric radioactivity levels in Finnish Lapland.



2. EXPERIMENTAL





**2.1 Sampling and procedures for the air filters before any chemical treatment**

The air filter samples were collected at Finnish Meteorological Institute's [FMI] Rovaniemi monitoring station (66°34´N, 25°50´E, elevation 198 m above sea level [a.s.l.]). Weekly sampled air volume was ~1000 $m^3$. First, total beta activity was measured from the filters five days after the end of sampling. Then the filters were combined to suitable sets for the gamma measurement and determination of $^{137}$Cs concentration. The details of air sampling, combining air filters and measurements for gamma activity of $^{137}$Cs and total beta activity have been given in Salminen et al. (2019).

**2.2 Radiochemical separation of Pu, Am, and Sr from air filters**

The detailed description about the radioanalytical separation procedure and the radionuclide measurements is given elsewhere (Salminen-Paatero and Paatero, submitted to MethodX). $^{238,239,240}$Pu, $^{241}$Am, and $^{90}$Sr were separated from dissolved air filter sample sets containing filters from three months to five years. The separation method included extraction chromatography and anion exchange steps and it was modified from original method for air filters of 1-3 days sampling time, presented in Salminen and Paatero (2009).

**2.3 Measurement of $^{238,239,240}$Pu, $^{241}$Am, $^{90}$Sr, and $^{240}$Pu/$^{239}$Pu in the air filter samples**

The activity concentration of alpha emitting Pu isotopes $^{238}$Pu and $^{239+240}$Pu in the air filter samples was determined by Alpha Analyst spectrometer (Canberra), the activity concentration of $^{90}$Sr by Quantulus 1220 liquid scintillation counter (LSC) via the activity concentration of the daughter nuclide $^{90}$Y. Finally, after an additional purification step of the Pu alpha counting samples, the mass ratio $^{240}$Pu/$^{239}$Pu was determined by SF-ICP-MS (Sector-Focusing Inductively Couple Plasma-Mass Spectrometry), ELEMENT XR (Thermo Scientific). More detailed description of the measurements is given in Salminen-Paatero and Paatero (submitted to MethodX).



91

## 3. RESULTS AND DISCUSSION

### 3.1 The activity concentrations of $^{238}$Pu, $^{239+240}$Pu, and $^{241}$Pu in the surface air of Rovaniemi in 1965-2011

95

*3.1.1 The activity concentration of $^{238}$Pu*

The activity concentration of $^{238}$Pu had the highest value of 259±13 nBq m$^{-3}$ in 1968 during the investigated time period 1965-2011 (Table 1, Fig. 1). The years of the highest concentrations of $^{238}$Pu around 1968 are a consequence from the destruction of SNAP-9A nuclear power unit of the satellite re-entering the atmosphere in 1964. Since 1968, the activity concentration of $^{238}$Pu in the surface air of Rovaniemi has been decreasing being nowadays below or close to the detection limit. The concentration of $^{238}$Pu was under detection limit also during the months after the Chernobyl accident, in April-December in 1986.

104

*3.1.2 The activity concentration of $^{239+240}$Pu*

The activity concentration of $^{239+240}$Pu in the surface air of Rovaniemi has been dropping from the highest value 2270±40 nBq m$^{-3}$ in 1965, being a few nBq m$^{-3}$ since 1996 (Table 1, Fig. 1). Two years before the sampling was started, in 1963, was the deposition maximum from atmospheric nuclear tests that were performed before the Partial Test Ban Treaty. For example, at Sodankylä, 120 km North of Rovaniemi the average $^{239+240}$Pu activity concentration was 17 000 nBq m$^{-3}$ in 1963 (Salminen & Paatero 2009). Slight peaks in the $^{239+240}$Pu concentration can be seen in 1974, 1978 and 1981, evidently due to the atmospheric nuclear tests performed by People's Republic of China between 1973 and 1980. The effect of these nuclear tests on the radionuclide concentration level in Finnish Lapland was already observed in the concentration variation of $^{137}$Cs (Salminen-Paatero et al. 2019). Like with $^{238}$Pu, the concentration of $^{239+240}$Pu was below the detection limit in April-June 1986 following the Chernobyl accident. For comparison, the concentration of $^{239+240}$Pu was 32 µBq m$^{-3}$ in the surface air in Nurmijärvi (Southern Finland), in 28 April, 1986 (Jaakkola et al. 1986).

Based on the extremely low activity concentrations of both $^{238}$Pu and $^{239+240}$Pu in the surface air of Rovaniemi during April-December 1986, one can conclude that hardly any plutonium was migrated to Finnish Lapland from the destroyed Chernobyl nuclear reactor after 26$^{th}$ April, 1986. This conclusion is also supported by high concentration of $^{137}$Cs (1294±7 µBq m$^{-3}$) and low concentration





of $^{90}$Sr (5.2±1.1 µBq m$^{-3}$) in the same air filter samples in April-June 1986 (Salminen-Paatero et al.
2019). It has been suggested that the initial contamination plume from the destroyed Chernobyl
reactor contained intermediate ($^{90}$Sr) and refractory elements (Pu isotopes) and that plume passed
over Central and Southern parts of Finland, while the volatile elements like $^{137}$Cs were mostly in the
later contamination plumes which reached also Lapland (Saxén et al. 1987). However, observations
of $^{241}$Pu/$^{239+240}$Pu activity ratio discussed in a later paragraph show some possibility of Chernobyl-
derived plutonium in Finnish Lapland.

### 130 *3.1.3 The activity concentration of $^{241}$Pu*

The concentration of $^{241}$Pu was calculated via ingrowth of $^{241}$Am and like with $^{239+240}$Pu, the activity
concentration of $^{241}$Pu had the highest value in 1965, 38 198±711 nBq m$^{-3}$, and since then its
concentration has been decreasing except small peaks in 1974, 1978, and 1981 (Table 1, Fig. 2).
Similarly with the activity concentration changes of $^{239+240}$Pu, these peaks in the activity concentration
of $^{241}$Pu are presumably caused by nuclear tests executed in People's Republic of China. The
atmospheric activity concentration of $^{241}$Pu was lower than the detection limit in April-June 1986,
and since July-December 1986, the amount of $^{241}$Pu was returned again to the same level as it was
before the Chernobyl accident in the surface air of Rovaniemi. Based on the $^{241}$Pu concentration only,
there is no evidence about any Chernobyl-derived $^{241}$Pu in Rovaniemi.
Interestingly, the increase in the activity concentration of $^{241}$Pu is seen in 2011, unlike with
$^{238,239,240}$Pu. The activity concentration of $^{241}$Pu in 2011, 602±131 nBq m$^{-3}$, is higher than the
concentration level in Rovaniemi during last decades before 2011, and it is probably due to the
Fukushima accident in 11$^{th}$ March 2011. The activity of $^{241}$Pu has been reported to be much higher
than the activity of $^{239+240}$Pu in the emissions from the destroyed Fukushima NPP, the activity ratio
$^{241}$Pu/$^{239+240}$Pu having a value of 108 in soil and litter samples (Zheng et al. 2012). The activity
concentrations of Pu isotopes were 25 000 nBq m$^{-3}$ for $^{241}$Pu, 130 nBq m$^{-3}$ for $^{239}$Pu and 150 nBq m$^{-3}$
$^{3}$ for $^{240}$Pu in the air filters sampled at 120 km from Fukushima on 15$^{th}$ March, 2011 (Shinonaga et al.

148 2014).

It is unfortunate that there is only one combined air filter sample from Rovaniemi for the year 2011,
because the annual concentration is only an average of the weekly concentrations in 2011 and now
the signal from the Fukushima accident has been diluted under excess effect of global fallout in the
air filters. It would have been interesting to analyze plutonium isotopes in weekly filters separately





from March 2011, for determining Fukushima-derived [241]Pu concentration and isotope ratios in
Finnish Lapland.


**157    3.2 The activity ratios [238]Pu/[239+240]Pu, [241]Pu/[239+240]Pu, [239+240]Pu/[137]Cs, total beta activity/[239+240]Pu,**

**158    and mass ratio [240]Pu/[239]Pu in the air filters**

*3.2.1 [238]Pu/[239+240]Pu activity ratio*
The activity ratio [238]Pu/[239+240]Pu was 0.022±0.003-0.444±0.023 in Rovaniemi in 1965-2011, the
values under the detection limit excluded (Table 2, Fig. 3). The variation in the activity ratio values
is 200-fold. The activity ratio [238]Pu/[239+240]Pu in the surface air can vary greatly even in a short time
due to e.g. stratospheric-tropospheric exchange, resuspension and introduction of several
contamination sources. For example, the activity ratio [238]Pu/[239+240]Pu varied from 0.014±0.003 to
0.32±0.11 in Sodankylä, Finnish Lapland, during one year in 1963, still the most typical value was
~0.03 that represents the activity ratio for the global fallout (Salminen and Paatero 2009). The ratio
started to increase in 1966 in Rovaniemi reaching a maximum in 1967 due to the previously
mentioned SNAP-9A satellite accident in 1964. Previously, an increased [238]Pu/[239+240]Pu activity ratio
due to the SNAP-9A accident has been found in lichens both in Subarctic Finland (Jaakkola et al.
1978) and Sweden (Holm and Persson 1975) a couple of years after 1964. This over two year delay
after the accident indicates how slow the interhemispheric transport of stratospheric radionuclides is
(Fabian et al. 1968).
The activity ratio [238]Pu/[239+240]Pu cannot be determined for the period immediately after the Chernobyl
accident because the activity concentrations of [238]Pu and [239+240]Pu were below the detection limit
during April-December 1986. This finding is in agreement with the previous assumptions about
hardly any Chernobyl-derived refractory elements in Finnish Lapland (Salminen-Paatero et al. 2019).
Due to the activity concentrations of [238]Pu and [239+240]Pu being below the detection limit, the activity
ratio [238]Pu/[239+240]Pu cannot be determined for the year of the Fukushima accident, 2011, either. For
comparison, both [238]Pu and [239+240]Pu were detected soon after the Fukushima accident in Lithuania,
~ 1300 km south from Rovaniemi (Lujanienė et al. 2012). The combined air filter sample set in
Lithuanian study contained the sampled air volume of ~2 x 10[6] m[3] during March 23 – April 15 2011,
the activity concentration of [239+240]Pu being 44.5±2.5 nBq m[-3], and the activity concentration of [238]Pu
being 1.2 times higher than of [239+240]Pu. The resulting activity ratio [238]Pu/[239+240]Pu in Lithuania was





1.2, clearly deviating from the activity ratio values in the Chernobyl fallout and global fallout from
nuclear weapons testing.

*3.2.2 $^{241}Pu/^{239+240}Pu$ activity ratio*
The activity ratio $^{241}Pu/^{239+240}Pu$ varied between 8.2±0.7 and 79±17 in the surface air of Rovaniemi
in 1965-2011, except April-December 1986 and 2011, when the concentration of one or both isotopes
(either $^{239+240}Pu$ or $^{241}Pu$) was under detection limit (Table 2, Fig. 4). These two periods following the
accidents of Chernobyl and Fukushima would have interesting $^{241}Pu/^{239+240}Pu$ activity ratio values for
determining the Pu contamination source in Rovaniemi. Unfortunately, the concentration of $^{239+240}Pu$
in the surface air of Finnish Lapland was extremely low during those periods.
The $^{241}Pu/^{239+240}Pu$ activity ratio values of Rovaniemi are mainly due to atmospheric nuclear weapons
testing in 1965-March 1986 and for the years 1987-2005, an influence from the Chernobyl accident
can be seen as elevated ratio values. The $^{241}Pu/^{239+240}Pu$ activity ratio was determined to be 15 in fresh
nuclear fallout in 1963-1972 (Perkins and Thomas 1980) and the corresponding ratio values in the
fallout from the Chernobyl accident have been 85 in Sweden and Poland (Holm et al. 1992; Mietelski
et al. 1999), and 95 in Finland (Paatero et al. 1994). The published $^{241}Pu/^{239+240}Pu$ activity ratio values
for the Fukushima-derived contamination are also high, e.g. 89 in air filters (calculated from the
individual isotope concentrations in Shinonaga et al. (2014)), and 108 in soil and litter samples (Zheng
et al. 2012).

*3.2.3 $^{240}Pu/^{239}Pu$ mass ratio*
The mass ratio $^{240}Pu/^{239}Pu$ was 0.117±0.009-0.278±0.093 in 1965-2011 (Table 2, Fig. 5) and the
major part of the ratio values correspond to the value ~0.18 for global fallout from atmospheric
nuclear weapons testing in Northern hemisphere (Beasley et al. 1998), taking into account the relative
measurement uncertainties. The highest mass ratio value occurred in April-June 1986, while the
activity concentrations of $^{238}Pu$, $^{239+240}Pu$ and $^{241}Pu$ were under detection limit by alpha spectrometry.
Therefore, it was possible to determine $^{239}Pu$ and $^{240}Pu$ by mass spectrometry even from the samples
with very low Pu-concentration (April-December 1986, 2011, etc.), although the relative
measurement uncertainties by ICP-MS are much higher for these samples with very low Pu-
concentration compared to the measurement uncertainties of samples with higher Pu-concentration
level.



The mass ratio $^{240}$Pu/$^{239}$Pu is higher in the emissions from the destroyed Chernobyl reactor, compared
to the global fallout value. For example, the mass ratio value 0.408±0.003 has been determined from
the samples of Chernobyl-contaminated soil layer (Muramatsu et al. 2000) and two hot particles that
migrated to Finland from Chernobyl had the mass ratios 0.33±0.07 and 0.53±0.03 (Salminen-Paatero
et al. 2012). The air filters sampled in Rovaniemi in April-June and July-December 1986 seem to
have elevated mass ratios, 0.278±0.093 and 0.254±0.073 respectively, but taking into account their
high measurement uncertainties, these post-Chernobyl ratio values might be close to the global fallout
ratio 0.18 after all.
Similarly with the refractory element emissions from the Chernobyl accident, the released fuel
particles from the Fukushima accident have significantly higher mass ratio $^{240}$Pu/$^{239}$Pu than the
global fallout value 0.18. Dunne et al. (2018) have compared the mass ratios $^{240}$Pu/$^{239}$Pu in soil,
sediment and vegetation samples collected at surroundings of Fukushima with the known mass
ratios in global fallout and in destroyed nuclear reactors of Fukushima NPP. The mass ratio
$^{240}$Pu/$^{239}$Pu for the Fukushima reactor units were obtained by using ORIGEN code, being 0.344 for
Reactor 1, 0.320 for Reactor 2, and 0.356 for Reactor 3, respectively (Nishihara et al. 2012). All
investigated environmental samples from proximity of Fukushima had the $^{240}$Pu/$^{239}$Pu atom ratios
between the global fallout value and the value for Reactor Unit 3 calculated by ORIGEN, with
exception of one deviating value (Dunne et al. 2018).
It was highlighted in the same study that the concentration level of Pu isotopes and the mass ratio
$^{240}$Pu/$^{239}$Pu varies greatly in the environment of Fukushima, and they don't necessarily correlate
with each other. Also the lowest mass ratio values in Fukushima have been at global fallout level.
This variety of isotope concentrations and isotope ratios has been noticed in other Fukushima-
related investigations as well. From a litter and soil sample set collected at 20-32 km from
Fukushima, three samples had high $^{241}$Pu concentrations and mass ratios 0.303-0.330 that can be
considered as representing contamination from the destroyed reactors of Fukushima (Zheng et al.
2012). The rest of the soil and litter samples from proximity of Fukushima in (Zheng et al. 2012)
had low $^{241}$Pu concentrations and the $^{240}$Pu/$^{239}$Pu mass ratios were at the Northern hemisphere
global fallout level. In another study, the air filter samples collected at 120 km from Fukushima
formed two groups: one having low $^{239}$Pu concentration and fairly similar mass ratio to global
fallout (0.141±0.002) and another having high $^{239}$Pu concentration and mass ratio clearly deviating
from global fallout (≥ 0.3) (Shinonaga et al. 2014).
The $^{240}$Pu/$^{239}$Pu mass ratio was only 0.145±0.091 in the surface air of Rovaniemi during the year of
the Fukushima accident, 2011. Again, the activity concentrations of both $^{239}$Pu and $^{240}$Pu were



extremely low in Rovaniemi during that year and the uncertainty of the mass ratio is therefore high,
suggesting that the ratio value in 2011 is probably due to global fallout though.

*3.2.4 $^{239+240}Pu/^{137}Cs$ activity ratio*
The activity ratio $^{239+240}Pu/^{137}Cs$ varied between 0.0005±0.0001 and 0.0393±0.0038 in the surface air
of Rovaniemi in 1965-2011, excluding the samples from April-December 1986 and 2011, when the
concentration of $^{239+240}Pu$ fell below the detection limit (Table 2). The lowest value for the activity
ratio occurred in 2006-2010, when the activity concentration of both radionuclides ($^{239+240}Pu$ and
$^{137}Cs$) has been constantly decreasing in the surface air for decades. The range of the values in
Rovaniemi is in agreement with previous studies of surface air in Finland. The activity ratio
$^{239+240}Pu/^{137}Cs$ was 0.0020±0.0008–0.029±0.010 in Sodankylä (Finnish Lapland) during 1963
(Salminen-Paatero and Paatero 2012) and 0.005±0.002–0.012±0.004 (range of annual mean values)
in Helsinki (Southern Finland) in 1962-1977 (Jaakkola et al. (1979).
Bossew et al. (2007) have calculated the reference values for the $^{239+240}Pu/^{137}Cs$ activity ratio in global
fallout and the Chernobyl accident, obtaining 0.0180±0.0024 (data from Bunzl and Kracke, 1988)
and 6.6 x 10$^{-6}$ (data from Irlweck and Khademi, 1993), respectively. The values of Rovaniemi are
higher than the value for Chernobyl contamination and some values of Rovaniemi are even higher
than the value for global fallout.
On the contrary to high $^{239+240}Pu/^{137}Cs$ ratio values in the surface air of Rovaniemi and in global
fallout, very low $^{239+240}Pu/^{137}Cs$ activity ratios have been observed in Fukushima environment.
Among all litter and soil samples from Fukushima in the study by Zheng et al. (2012), the three
samples that represent the Fukushima-derived contamination, i.e. have both high $^{241}Pu$ concentration
and high $^{240}Pu/^{239}Pu$ mass ratio, had the $^{137}Cs/ ^{239+240}Pu$ activity ratios 4 x 10$^{-8}$, 2 x 10$^{-7}$, and 5 x 10$^{-6}$
in 2011.

*3.2.5 Total beta activity/$^{239+240}Pu$ activity ratio*
The ratio between total beta activity (Salminen-Paatero et al. 2019) and $^{239+240}Pu$ remains rather
constant during the atmospheric nuclear testing era (Fig. 6). The ratio reflects the produced nuclide
composition after fission and activation reactions in the detonating devices. Following the Chernobyl
accident, the ratio increases almost three orders of magnitude. After the initial explosion plume, the
emissions from the burning reactor were dominated by volatile fission products, which explains the





high total beta activity/$^{239+240}$Pu activity ratio. After the decay of short-lived fission products, the ratio
soon returns close to the pre-Chernobyl level. Towards the end of the 20[th] century, the ratio starts
gradually increasing. This is explained by the decreasing amount of plutonium in the atmosphere
while the total beta activity remains on a constant level due to natural atmospheric radioactivity,
mainly $^{210}$Pb.

**3.3 Effect of actual and hypothetic nuclear detonations on the surface air of Finnish subarctic**

At least two new nuclear facilities in or close to the Euroarctic region are under preparation. A
construction of infrastructure for a new nuclear power plant at Pyhäjoki, western Finland, has been
started. Shtokmann natural gas field is located in the Barents Sea about halfway between northern
Finland and Novaya Zemlya. The future production facility has been planned to be powered by a
floating nuclear power plant. In case of hypothetical accidents in these plants, the atmospheric
dispersion of plutonium contamination was assessed with atmospheric transport modeling.
$^{241}$Pu dispersion in the atmosphere was simulated with the SILAM model (Sofiev et al., 2006; 2008).
The model runs were based on the meteorological forecast data of the European Centre for Medium-
Range Weather Forecasts (www.ecmwf.int) with a horizontal resolution of 0.25 degrees and with 9
vertical levels up to the height of 7700 m. Transport and dispersion calculations for both sites were
made for each day in the year 2010. Average activity concentrations of $^{241}$Pu in the surface air during
the first 48 hours after the release were calculated.
The following accident conditions, previously listed in Paatero et al. (2014), for the Pyhäjoki power
reactor (64°32'N, 24°15'E) were used:
-   pressurized water reactor with a thermal power of 4000 MW,
-   the end of the refueling interval,
-   an immediate release after shutdown with an effective release height of 200 m above
sea level, and
-   a $^{241}$Pu inventory of 6.2x10$^{17}$ Bq, release fraction of 0.1%, and a release of 6.2x10$^{14}$ Bq.

The following accident conditions for the case of Shtokmann gas field, Barents Sea (73°N, 44°E)
were used (previously used by Paatero et al. 2014):





-       ice breaker reactor with a fuel burnup of 466000 MWdays T$^{-1}$ HM,
-       an immediate release two hours after shutdown,
-       a radionuclide inventory according to Reistad and Ølgaard (2006),
-       an effective release height of 100 m above sea level, and
-       a $^{241}$Pu inventory of $3.2 \times 10^{14}$ Bq, release fraction of 0.2%, and a release of $6.4 \times 10^{11}$ Bq.

Varying meteorological situations have a decisive effect on the atmospheric plutonium transport
following accidental emissions from a nuclear reactor. The wind direction determines the path of the
emission plume. The wind speed sets how quickly the emission plume is advected. However, the
wind speed also affects the turbulence that disperses the plume vertically and horizontally. This
influences the plutonium concentrations in the air. Precipitation, for one's part, efficiently scavenges
plutonium-bearing particles from the atmosphere to the surface, which affects the deposition of
plutonium and furthermore its transfer to food webs.
From the Rovaniemi region point of view the worst of the calculated 365 dispersion cases would have
caused in ground-level air an average $^{241}$Pu activity concentration less than 1 kBq m$^{-3}$ during the first
48 hours after the release (Fig. 7). This equals to an annual average $^{241}$Pu exposure of 5 Bq m$^{-3}$. For
comparison, due to the atmospheric nuclear tests the $^{241}$Pu activity concentration varied between a
few dozens and some 1700 µBq m$^{-3}$ in 1963 in northern Finland, in other words on a several orders
of magnitude lower level (Salminen and Paatero 2009). In practice, the human exposure to $^{241}$Pu *via*
inhalation would remain on a clearly lower level because the civil defence authorities would order
the population to stay indoors with the ventilation systems closed and doors and windows sealed.
Compared to the Pyhäjoki accident scenario the consequences after a hypothetical accident in a
floating nuclear reactor in the Barents Sea would remain much less significant from the northern
Finland point of view. This is due to the smaller emissions, greater distance and favorable climatic
conditions, namely prevailing wind directions from the west and south-west. Only one case out of
365 dispersion calculations produced an atmospheric transport pattern that reached northernmost
Finland (Fig. 8). The ground-level $^{241}$Pu activity concentrations would have been less than 0.01 Bq
m$^{-3}$ during the first 48 hours corresponding to an annual average concentration of 55 µBq m$^{-3}$. This is
similar to the activity concentrations occurring in the early 1960s.




### 3.4 Case "Fukushima 2011 and $^{241}$Pu"

In an earlier work by Paatero et al. (2012), it was observed that the Silam model simulates well the temporal behavior of the Fukushima emission plume in the High Arctic. The calculated activity concentration levels, however, were an order of magnitude lower than the observed ones. This deviation was attributed to the inaccuracies in the source term. From the same model dataset the $^{137}$Cs activity concentration in the surface of Rovaniemi was extracted. The level of these values was then corrected by adjusting them to the observed weekly $^{137}$Cs activity concentration of 170 µBq m$^{-3}$ between 28 March and 4 April 2011 (Salminen-Paatero et al. 2019). From these values the $^{241}$Pu activity concentrations were obtained by multiplying with the $^{241}$Pu/$^{137}$Pu activity ratio of $7.81 \times 10^{-6}$. This activity ratio was found in hot particles close to the Fukushima Daiichi NPP by Igarashi et al. (2019). The calculated hourly $^{241}$Pu activity concentrations reach a maximum level of 0.01 µBq m$^{-3}$ for two short periods (Fig. 9). The calculated peak activity concentrations are very low, six orders of magnitude, compared with daily $^{241}$Pu activity concentrations observed in northern Finland in 1963 (Salminen and Paatero 2009). However, there is a discrepancy between this assessment and the annual observed $^{241}$Pu activity concentration of 0.6 µBq m$^{-3}$ (Fig. 2). If we assume that the background $^{241}$Pu activity concentration due to the atmospheric nuclear tests and the Chernobyl accident would be 0.03 µBq m$^{-3}$ then the average activity concentration between 27 March and 17 April should be 9.3 µBq m$^{-3}$, in other words a thousand times higher. An obvious explanation is that the $^{241}$Pu/$^{137}$Pu activity ratio ($7.81 \times 10^{-6}$) we used is not valid. The value may not be representative to bulk emission mixture of the destroyed reactors. Zheng et al. (2012) found out that the $^{137}$Cs/$^{239,240}$Pu activity ratio in environmental samples varied over four orders of magnitude. In addition, the hot particles were found close to the source and fractionation processes during the over 10000 km long atmospheric transport could occur too.

### 4. CONCLUSIONS

Based on the activity concentrations of $^{238,239,240,241}$Pu, hardly any refractory elements from the exploded Chernobyl reactor reached Finnish Lapland in 1986. Previously Chernobyl-derived $^{137}$Cs,



a more volatile isotope, has been detected from the same air filter samples whereas there was no
increased concentration of $^{90}$Sr in the samples after March 1986. The influence from the Fukushima
Daiichi accident is seen as increased concentration of $^{241}$Pu in the air filters. Nuclear weapons testing
in 1950's and 1960's, later nuclear tests in 1973-1980, SNAP 9A-satellite accident in 1964, and the
Fukushima accident in 2011 have been the main sources of Pu in the surface air in Finnish Lapland
during 1965-2011.
Overall, the mass ratio $^{240}$Pu/$^{239}$Pu is more sensitive contamination source indicator than the activity
ratios $^{238}$Pu/$^{239+240}$Pu or $^{241}$Pu/$^{239+240}$Pu due to lower detection limit of ICP-MS compared to alpha
spectrometry and LSC. However, it is always useful to analyze more than one isotope ratio or activity
ratio, and single isotope concentrations when characterizing the origin of Pu contamination. In this
case, the contribution of the Fukushima accident in Rovaniemi would not have been observed without
analyzing the concentration of $^{241}$Pu in the air filter samples.
Dispersion modelling results with the atmospheric dispersion model Silam indicate that Pu
contamination in northern Finland would be negligible due to a hypothetical accident in a floating
nuclear reactor at the Shtokmann natural gas field, Barents Sea. The Pu contamination risk would be
higher in case of a severe accident at the intended nuclear power plant at Pyhäjoki, western Finland,
due to the bigger reactor and shorter distance.



ACKNOWLEDGEMENTS
Emil Pesonen is acknowledged for help with cutting the air filter samples for ashing and Ilia
Rodushkin (ALS Scandinavia Luleå laboratory) for measuring the Pu samples with ICP-MS. This
work belongs to "Collaboration Network on EuroArctic Environmental Radiation Protection and
Research (CEEPRA)". The project was funded by EU Kolarctic ENPI CBC 2007-2013 programme
that was managed by the Regional Council of Lapland. The authors want to thank EU-project
''TOXI Triage'' (Project id. 653409) for additional support.

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





Table 1. The atmospheric activity concentrations of $^{238}$Pu, $^{239+240}$Pu and $^{241}$Pu in Rovaniemi, Finnish
Lapland. The activity values have been decay-corrected to the middle point of the sampling period.

| Year | A $^{238}$Pu (nBq m$^{-3}$) | A $^{239+240}$Pu (nBq m$^{-3}$) | A $^{241}$Pu (nBq m$^{-3}$) |
|---|---|---|---|
| 1965 | 68±8 | 2270±40 | 38198±711 |
| 1966 | 117±7 | 1371±21 | 21182±451 |
| 1967 | 221±10 | 497±13 | 7768±236 |
| 1968 | 259±13 | 969±20 | 16237±396 |
| 1969 | 245±12 | 973±20 | 14585±372 |
| 1970 | 135±9 | 1040±20 | 15027±367 |
| 1971 | 76±5 | 1211±16 | 15975±387 |
| 1972 | 28±3 | 325±7 | 3456±179 |
| 1973 | 26±3 | 206±7 | 1701±128 |
| 1974 | 13±2 | 570±12 | 7383±261 |
| 1975 | 15±3 | 250±10 | 3769±182 |
| 1976 | 6.7±1.2 | 74±3 | 804±75 |
| 1977 | 6.9±1.2 | 297±7 | 3632±169 |
| 1978 | 13±2 | 563±10 | 9106±291 |
| 1979 | 6.1±1.2 | 175±5 | 3645±210 |
| 1980 | 2.7±0.9 | 74±4 | 1063±92 |
| 1981 | 7.0±1.7 | 248±9 | 2137±137 |
| 1982-March 1986 | 0.59±0.16 | 15.3±0.8 | 200±19 |
| April-June 1986 | < 1.6 | < 7.2 | < 381 |
| July-December 1986 | < 1.1 | < 5.2 | 315±71 |
| 1987-1990 | 2.2±0.3 | 5.8±0.4 | 101±15 |
| 1991-1995 | 0.23±0.07 | 16.9±0.1 | 73±11 |
| 1996-2000 | < 0.1 | 6.5±0.2 | 39±8 |
| 2001-2005 | 0.37±0.19 | 1.4±0.3 | 41±10 |
| 2006-2010 | < 0.4 | 0.51±0.14 | < 25 |
| 2011 | < 1.5 | < 3.5 | 602±131 |
















Table 2. The activity ratios $^{238}$Pu/$^{239+240}$Pu, $^{241}$Pu/$^{239+240}$Pu, $^{239+240}$Pu/$^{137}$Cs, and the mass ratio
$^{240}$Pu/$^{239}$Pu in the air filters collected in Rovaniemi. The uncertainty is 1 sigma error for the activity
ratios and 2 sigma error for the mass ratio. – means that one or both isotopes had concentration
below the detection limit.

| Year | A $^{238}$Pu/ A $^{239+240}$Pu | A $^{241}$Pu / A $^{239+240}$Pu | mass ratio $^{240}$Pu/$^{239}$Pu | A $^{239+240}$Pu/A $^{137}$Cs |
|---|---|---|---|---|
| 1965 | 0.030±0.004 | 16.8±0.4 | 0.177±0.006 | 0.0071±0.0001 |
| 1966 | 0.085±0.005 | 15.5±0.4 | 0.172±0.003 | 0.0067±0.0001 |
| 1967 | 0.444±0.023 | 15.6±0.6 | 0.170±0.003 | 0.0079±0.0003 |
| 1968 | 0.267±0.014 | 16.8±0.5 | 0.190±0.004 | 0.0108±0.0003 |
| 1969 | 0.252±0.014 | 15.0±0.5 | 0.172±0.005 | 0.0104±0.0003 |
| 1970 | 0.130±0.009 | 14.5±0.5 | 0.186±0.007 | 0.0087±0.0002 |
| 1971 | 0.063±0.004 | 13.2±0.4 | 0.174±0.006 | 0.0135±0.0002 |
| 1972 | 0.087±0.008 | 10.6±0.6 | 0.125±0.007 | 0.0116±0.0005 |
| 1973 | 0.125±0.015 | 8.2±0.7 | 0.131±0.008 | 0.0182±0.0009 |
| 1974 | 0.022±0.003 | 12.9±0.5 | 0.182±0.005 | 0.0102±0.0003 |
| 1975 | 0.058±0.011 | 15.1±0.9 | 0.132±0.008 | 0.0102±0.0005 |
| 1976 | 0.091±0.016 | 10.9±1.1 | 0.138±0.009 | 0.0130±0.0010 |
| 1977 | 0.023±0.004 | 12.2±0.6 | 0.216±0.015 | 0.0097±0.0004 |
| 1978 | 0.024±0.003 | 16.2±0.6 | 0.209±0.011 | 0.0102±0.0003 |
| 1979 | 0.035±0.007 | 20.8±1.4 | 0.209±0.012 | 0.0107±0.0004 |
| 1980 | 0.036±0.012 | 14.3±1.5 | 0.173±0.015 | 0.0090±0.0006 |
| 1981 | 0.028±0.007 | 8.6±0.6 | 0.117±0.009 | 0.0107±0.0005 |
| 1982-March 1986 | 0.038±0.011 | 13.1±1.4 | 0.142±0.011 | 0.0065±0.0006 |
| April-June 1986 | - | - | 0.278±0.093 | - |
| July-December 1986 | - | - | 0.254±0.073 | - |
| 1987-1990 | 0.376±0.056 | 18±3 | 0.152±0.026 | 0.0014±0.0001 |
| 1991-1995 | 0.245±0.082 | 79±17 | 0.132±0.091 | 0.0393±0.0038 |
| 1996-2000 | - | 32±8 | 0.131±0.066 | 0.0106±0.0010 |
| 2001-2005 | 0.260±0.142 | 29±9 | 0.170±0.082 | 0.0030±0.0007 |
| 2006-2010 | - | - | 0.194±0.116 | 0.0005±0.0001 |
| 2011 | - | - | 0.145±0.091 | - |







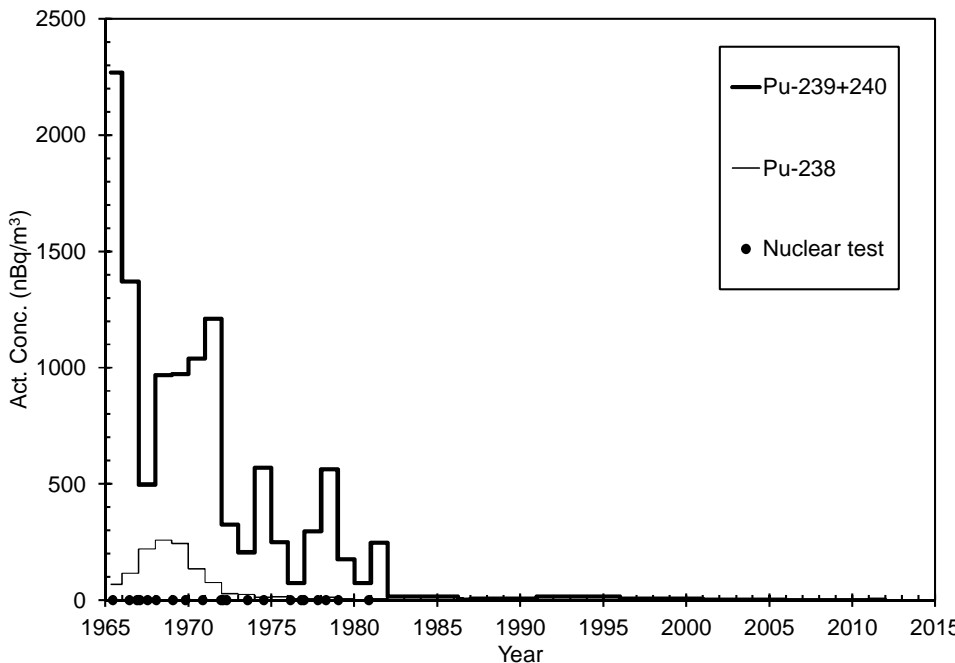


Fig.1. Activity concentration of $^{238}$Pu (thin line, nBq m$^{-3}$) and $^{239+240}$Pu (thick line, nBq m$^{-3}$) in surface air of Rovaniemi in 1965-2011. Values below the detection limit have been depicted as half the MDA value (Table 1). The black circles indicate the times of atmospheric nuclear tests in the People's Republic of China (UNSCEAR 2000).
















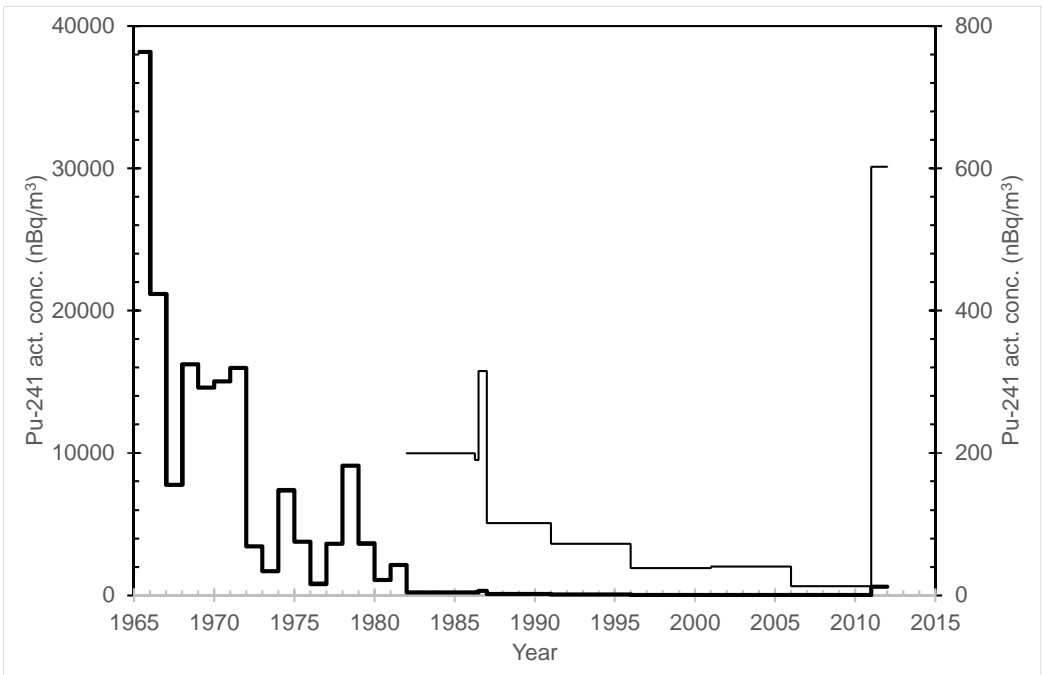


Fig. 2. Activity concentration of $^{241}$Pu (nBq m$^{-3}$) in surface air of Rovaniemi (thick line 1965-2011, left vertical scale; thin line 1982-2011, right vertical scale). Values below the detection limit have been depicted as half the MDA value (Table 1).




















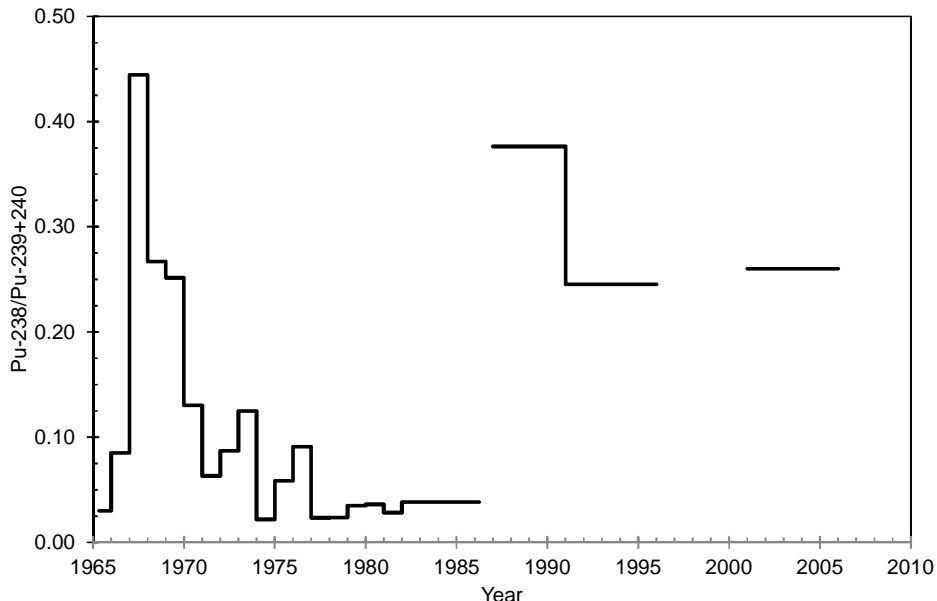


Fig. 3. The activity ratio $^{238}$Pu/$^{239+240}$Pu in surface air of Rovaniemi as a function of time.


















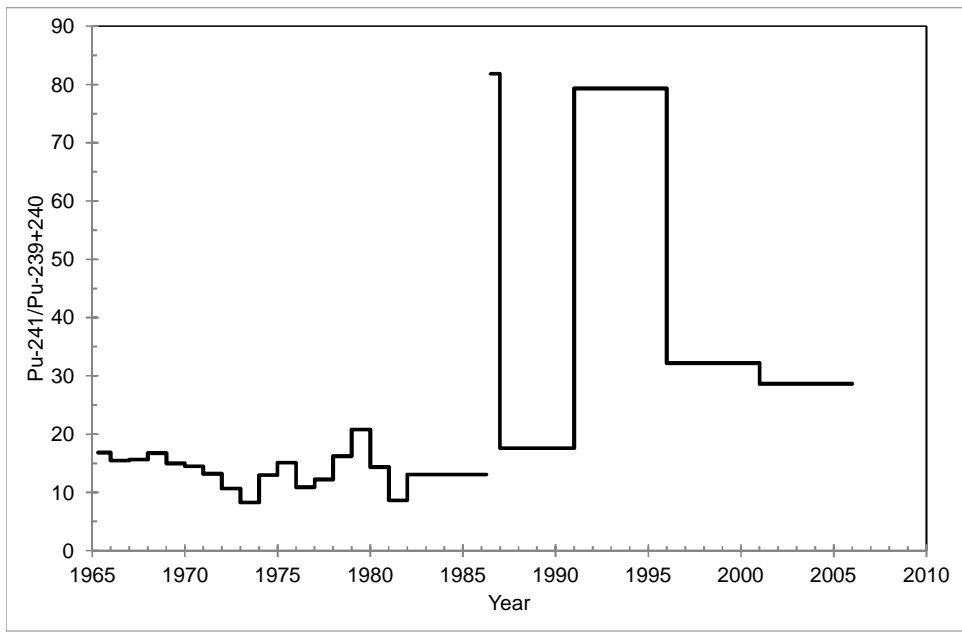


Fig. 4. The activity ratio $^{241}$Pu/$^{239+240}$Pu in surface air of Rovaniemi as a function of time.





















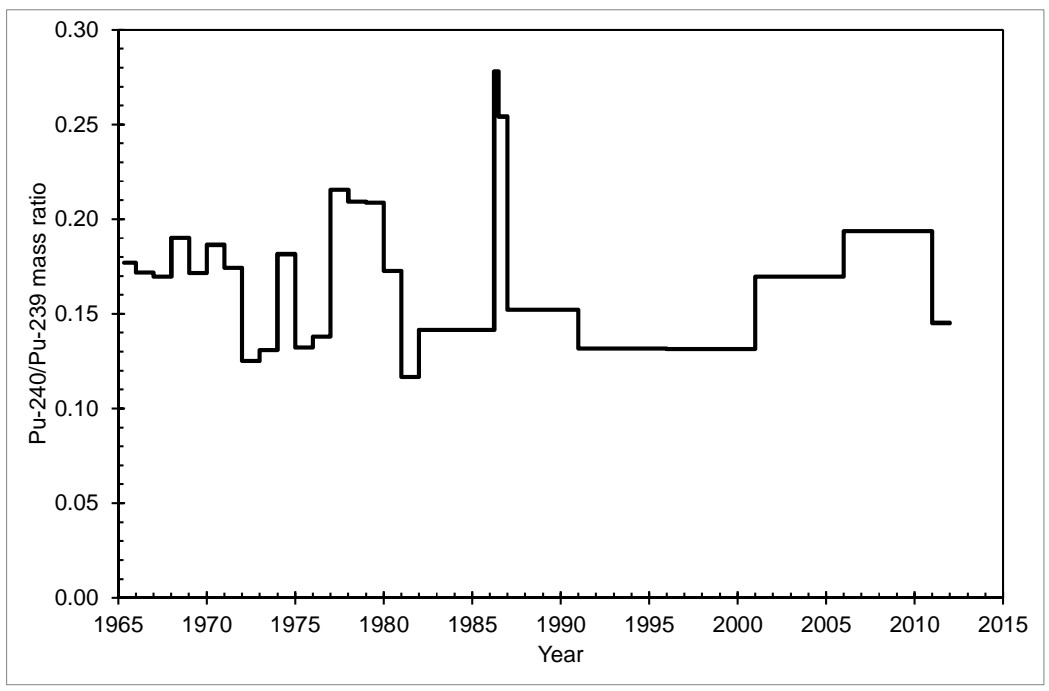


Fig 5. The mass ratio $^{240}$Pu/$^{239}$Pu in surface air of Rovaniemi as a function of time.















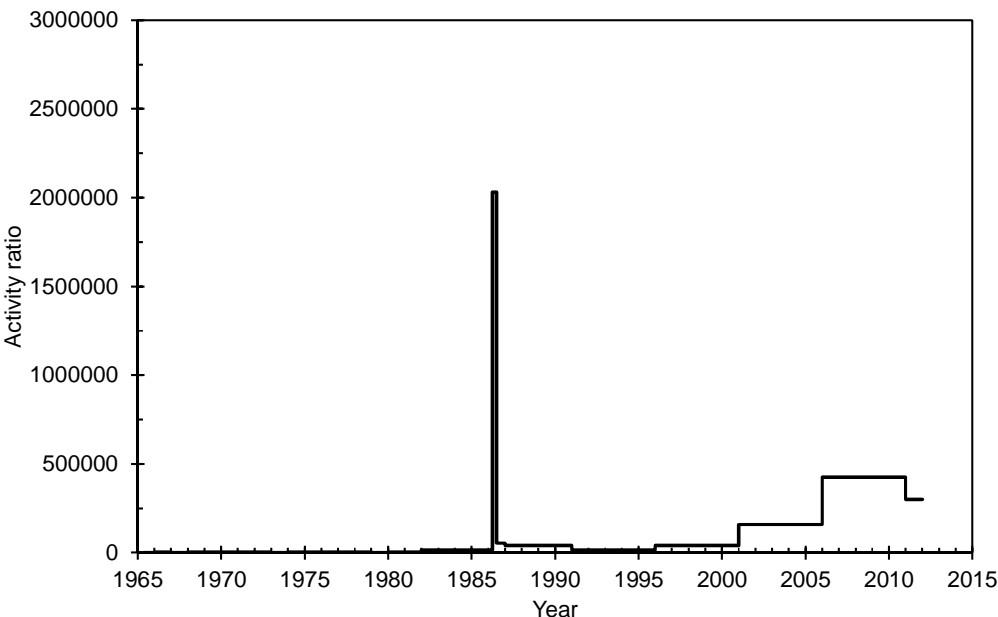


Fig. 6. The ratio of total beta activity (Salminen-Paatero et al. 2019) and $^{239+240}$Pu activity content of
surface air in Rovaniemi in 1965-2011. $^{239+240}$Pu values below the detection limit have been
replaced with half the MDA values in the ratio calculation (Table 1).



















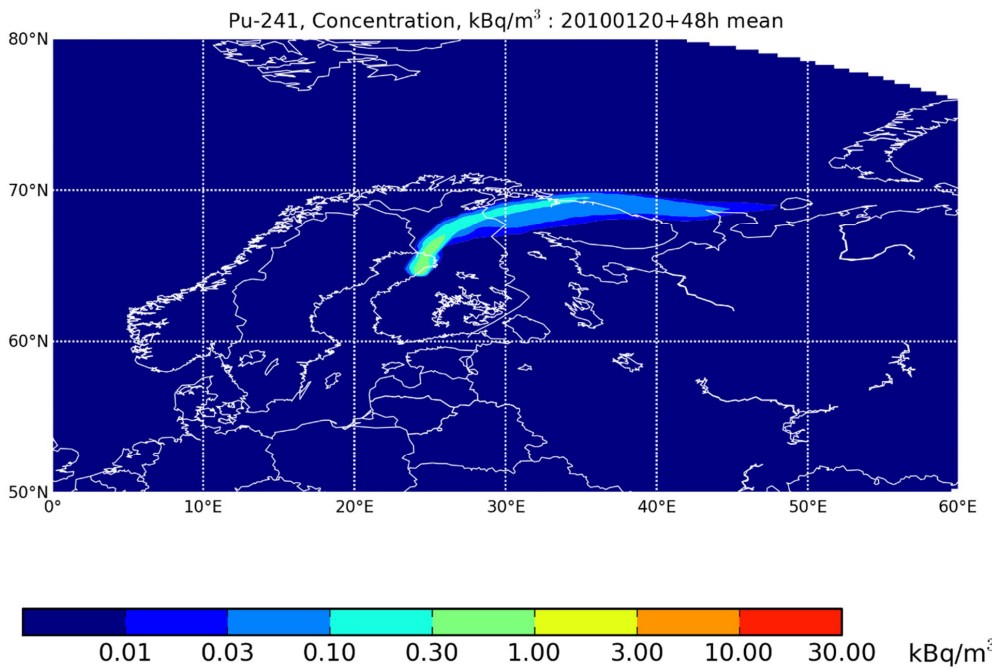


Fig. 7. Average activity concentration of $^{241}$Pu in the surface air during the first 48 hours after a
hypothetical reactor accident at Pyhäjoki, assumed release 20 January 2010.









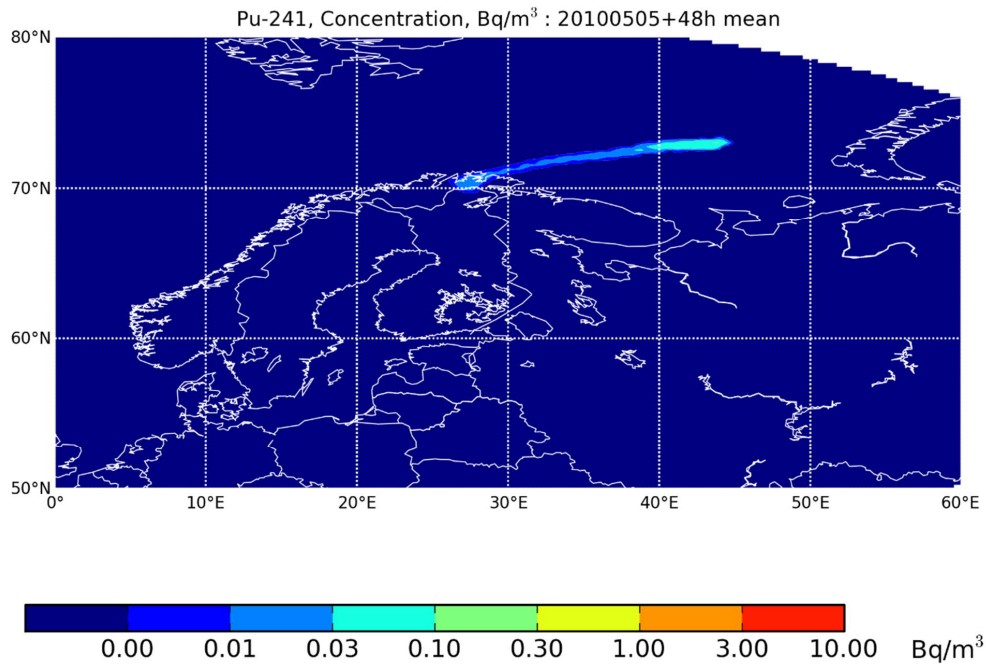


Fig. 8. Average activity concentration of [241]Pu in the surface air during the first 48 hours after a
hypothetical accident in a floating reactor at Shtokmann natural gas field, Barents Sea, assumed
release 5 May 2010.






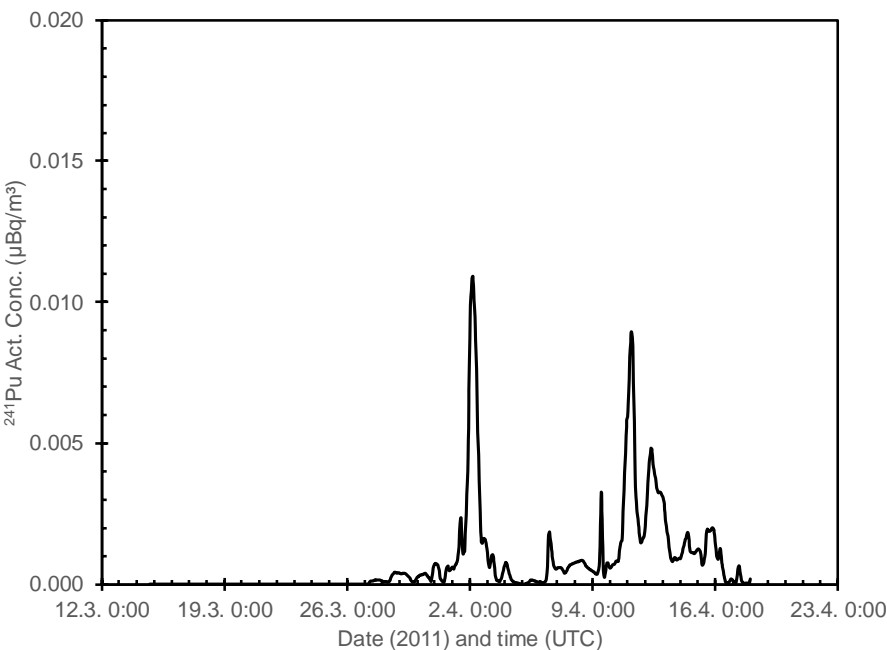


Fig. 9. Modeled hourly $^{241}$Pu activity concentration ($\mu$Bq m$^{-3}$) in surface air of Rovaniemi in March-
April 2011.






Author contributions
Susanna Salminen-Paatero performed radiochemical analysis and data analysis. Julius Vira
produced Silam calculations. Jussi Paatero provided the air filter sampling and sampling data, and
planned the accident scenarios. All authors contributed to writing the manuscript.

Data availability
Data will be available at University of Helsinki open data system.










