# Peer review of "Measurements and modeling of airborne plutonium in Subarctic Finland"

_Atmospheric Chemistry and Physics, 2019_

## Referee Comment (RC1) · Anonymous Referee #1 · 2 Jan 2020

**1    General Comments:**

The manuscript lists concentrations of radionuclides and isotope ratios sampled at Rovaniemi in Finnish Lapland between 1965 and 2011, and reports on daily 48 hour-duration radionuclide dispersion simulations from hypothetical accidents at planned nuclear power plants (NPP) over one year (2010) using the SILAM model.

Overall, the manuscript does not represent a substantial contribution to scientific progress (there are no substantial new concepts, ideas, or methods). In particular, the model simulations of potential NPP at specific locations fall outside the scope of

[Figure]

Atmospheric Chemistry and Physics. In my view the modelling part of the manuscript should be eliminated. The observations are discussed in a more balanced way (with consideration of related work, including appropriate references).

The two parts (observations and model) are disjoint and in particular the modelling component is not motivated scientifically and the description of calculations is not sufficiently complete and precise to allow their reproduction as the model setup and outcomes are not discussed in detail.

The presentation quality of the manuscript (in particular the use of the English language but also the quality of figures and tables) is not of the standard required for publication in ACP.

**2  Specific Comments:**

The abstract provides a concise and complete summary.

I propose that the tables listing concentrations are moved to a Supplement, as they are not directly referenced and their inclusion along with the timeline plots in Figures is superfluous.

**3  Techical Corrections:**

There are numerous editorial corrections required to reach publication standard, the authors should carefully follow the ACP guide for authors in editing the manuscript before re-submission.

---

## Author Comment (AC1) · 5 Jan 2020

We thank the reviewer for her/his careful reading and constructive criticism towards our work. As a consequence of these comments, the forthcoming modifications of the manuscript will improve greatly the quality of the final version.

General Comments: The manuscript lists concentrations of radionuclides and isotope ratios sampled at Rovaniemi in Finnish Lapland between 1965 and 2011, and reports on daily 48 hour-duration radionuclide dispersion simulations from hypothetical accidents at planned nuclear power plants (NPP) over one year (2010) using the SILAM model. Overall, the manuscript does not represent a substantial contribution to scien-

tific progress (there are no substantial new concepts, ideas, or methods).

Response: Only few long time series of atmospheric radioactivity exist from Subarctic and Arctic regions, and most of the existing time series contain only gamma emitters or fission products 137Cs and 90Sr. Producing atmospheric data of Pu isotopes is more laborious as they need to be radiochemically separated from the air filter matrix prior to activity measurement or isotope ratio determination. However, Pu isotope ratios provide important information about the nuclear contamination source in Subarctic and Arctic areas as they act like fingerprints in contamination identification. Finland is one exceptional example of significant unevenness of atmospheric nuclear contamination across a single state, since as was found out in this study, the large northern part of the country was mostly saved from the Chernobyl-derived transuranium deposition while the central and southern parts were more or less contaminated by the Chernobyl accident. The presence of plutonium isotopes in the air of high northern latitudes after the Fukushima accident has not been studied either. We see that due to all the listed reasons, it is meaningful to publish these results, preferably in ACP, and they will complete other observations and studies of plutonium sources and atmospheric contamination level in northern latitudes. Obviously, we have not expressed these justifications clearly enough in the current Introduction part and they will be now summarised to the Introduction for the revised version.

In particular, the model simulations of potential NPP at specific locations fall outside the scope of Atmospheric Chemistry and Physics. In my view the modelling part of the manuscript should be eliminated. The observations are discussed in a more balanced way (with consideration of related work, including appropriate references). The two parts (observations and model) are disjoint and in particular the modelling component is not motivated scientifically and the description of calculations is not sufficiently complete and precise to allow their reproduction as the model setup and out-comes are not discussed in detail.

Response: We believe that the modeling part of the manuscript completes the obser-

vational part, because 1) it provides risk estimates and reference contamination levels related to future nuclear activities in and close to Arctic regions that can be compared to earlier actual releases, and 2) it shows with the Fukushima case how important accurate information on the source term is for the prediction of resulting activity concentrations in the air following an atmospheric release of radioactivity. We will add a note on this justification to the manuscript. What comes to the description of calculations we have given the appropriate literature references concerning the model, the source of the meteorological data and the release parameters. We think that with the information provided, the dispersion calculations can be repeated with any similar computer models.

The presentation quality of the manuscript (in particular the use of the English language but also the quality of figures and tables) is not of the standard required for publication in ACP.

Response: Language re-check will be performed for the revised version as well as all figures and tables will be edited for the final published version according to the requirements by the editorial office.

2 Specific Comments: The abstract provides a concise and complete summary. I propose that the tables listing concentrations are moved to a Supplement, as they are not directly referenced and their inclusion along with the timeline plots in Figures is superfluous.

Response: In Results & Discussion part, both tables have been referenced several times in case of each nuclide and activity or mass ratio. However, the tables can be published as a separate Supplement part, if the Editor agrees with this modification.

3 Technical Corrections: There are numerous editorial corrections required to reach publication standard, the authors should carefully follow the ACP guide for authors in editing the manuscript before re-submission.

Response: Any typos or expressions not in line with the ACP manuscript format will be corrected first by the authors and eventually by the editorial office before publishing the final form.

---

## Referee Comment (RC2) · Anonymous Referee #2 · 2 Mar 2020

Of whole text my doubts arises only regarding the Pu-241 measurements observed for Fukushima time. From what is said in the manuscript, it remains for me unclear how exactly and when the results for Pu-241 in the samples from the year 2011 were obtained? In case of Pu-241 determination by ingrown of Am-241 the whole technical details of history of sample are important. What is obvious to me is that the Authors noticed the presence of 5.5 MeV alpha peak, which they did not attribute to Pu-238, what suggests, that samples were measured twice and 5.5 MeV peak was smaller in first measurement. The results from first measurement were used for determination of Pu-238 and after some years the same Pu source was re-measured and change of count rate in 5.5 MeV peak was interepreted as Am-241. Only the sampling time

points to Fukushima as origin. I will by much happy with the text if all those technical data (i.e. when Pu was separated from Am/Cm and when measured and when re-measured) data will appeaer in a small paragraph concering Pu-241 measurements in "Experimental" section. Please note, that in fresh spent nuclear fuel the main actinide alpha activity comes from Cm-242 (T1/2=160 days) and it is a bit striking, that this isotope was not detected along with Pu in reports on finding the Pu of Fukushima origin at the distances of 10 000 km. The lack of Cm-242 suggests something else then fresh release from nuclear reactor.

Besides my doubts on Pu-241 (or rather Am-241) presence interpretation as Fukushima origin I like whole manuscript. The first part present history of contaminations and the modelling part compare the past events with scenarios of possible future accidents. I think such concept is logical and answers to questions which comes from fears on possible nuclear accidents.

---

## Author Response (AR1)

**(1) Comments from referees/public**

Anonymous Referee #1

General Comments: The manuscript lists concentrations of radionuclides and isotope ratios sampled at Rovaniemi in Finnish Lapland between 1965 and 2011, and reports on daily 48 hour-duration radionuclide dispersion simulations from hypothetical accidents at planned nuclear power plants (NPP) over one year (2010) using the SILAM model. Overall, the manuscript does not represent a substantial contribution to scientific progress (there are no substantial new concepts, ideas, or methods). In particular, the model simulations of potential NPP at specific locations fall outside the scope of Atmospheric Chemistry and Physics. In my view the modelling part of the manuscript should be eliminated. The observations are discussed in a more balanced way (with consideration of related work, including appropriate references). The two parts (observations and model) are disjoint and in particular the modelling component is not motivated scientifically and the description of calculations is not sufficiently complete and precise to allow their reproduction as the model setup and outcomes are not discussed in detail. The presentation quality of the manuscript (in particular the use of the English language but also the quality of figures and tables) is not of the standard required for publication in ACP.

Specific Comments: The abstract provides a concise and complete summary. I propose that the tables listing concentrations are moved to a Supplement, as they are not directly referenced and their inclusion along with the timeline plots in Figures is superfluous.

Technical Corrections: There are numerous editorial corrections required to reach publication standard, the authors should carefully follow the ACP guide for authors in editing the manuscript before re-submission.

Anonymous Referee #2

Of whole text my doubts arises only regarding the Pu-241 measurements observed for Fukushima time. From what is said in the manuscript, it remains for me unclear how exactly and when the results for Pu-241 in the samples from the year 2011 were obtained? In case of Pu-241 determination by ingrown of Am-241 the whole technical details of history of sample are important. What is obvious to me is that the Authors noticed the presence of 5.5 MeV alpha peak, which they did not attribute to Pu-238, what suggests, that samples were measured twice and 5.5 MeV peak was smaller in first measurement. The results from first measurement were used for determination of Pu-238 and after some years the same Pu source was re-measured and change of count rate in 5.5 MeV peak was interpreted as Am-241. Only the sampling time points to Fukushima as origin. I will by much happy with the text if all those technical data (i.e. when Pu was separated from Am/Cm and when measured and when re-measured) data will appear in a small paragraph concerning Pu-241 measurements in "Experimental" section. Please note, that in fresh spent nuclear fuel the main actinide alpha activity comes from Cm-242 (T1/2=160 days) and it is a bit striking, that this isotope was not detected along with Pu in reports on finding the Pu of Fukushima origin at the distances of 10 000 km. The lack of Cm-242 suggests something else then fresh release from nuclear reactor.

Besides my doubts on Pu-241 (or rather Am-241) presence interpretation as Fukushima origin I like whole manuscript. The first part present history of contaminations and the modelling part compare the past events with scenarios of possible future accidents. I think such concept is logical and answers to questions which comes from fears on possible nuclear accidents.

**(2) Author's response**

**Response to R1 reviewer comments**

We thank the reviewer for her/his careful reading and constructive criticism towards our work. As a consequence of these comments, the forthcoming modifications of the manuscript will improve greatly the quality of the final version.

*General Comments: The manuscript lists concentrations of radionuclides and isotope ratios sampled at Rovaniemi in Finnish Lapland between 1965 and 2011, and reports on daily 48 hour-duration radionuclide dispersion simulations from hypothetical accidents at planned nuclear power plants (NPP) over one year (2010) using the SILAM model. Overall, the manuscript does not represent a substantial contribution to scientific progress (there are no substantial new concepts, ideas, or methods).*

Only few long time series of atmospheric radioactivity exist from Subarctic and Arctic regions, and most of the existing time series contain only gamma emitters or fission products 137Cs and 90Sr. Producing atmospheric data of Pu isotopes is more laborious as they need to be radiochemically separated from the air filter matrix prior to activity measurement or isotope ratio determination. However, Pu isotope ratios provide important information about the nuclear contamination source in Subarctic and Arctic areas as they act like fingerprints in contamination identification. Finland is one exceptional example of significant unevenness of atmospheric nuclear contamination across a single state, since as was found out in this study, the large northern part of the country was mostly saved from the Chernobyl-derived transuranium deposition while the central and southern parts were more or less contaminated by the Chernobyl accident. The presence of plutonium isotopes in the air of high northern latitudes after the Fukushima accident has not been studied either. We see that due to all the listed reasons, it is meaningful to publish these results, preferably in ACP, and they will complete other observations and studies of plutonium sources and atmospheric contamination level in northern latitudes. Obviously, we have not expressed these justifications clearly enough in the current Introduction part and they have been now summarised in the Introduction of the revised version.

*In particular, the model simulations of potential NPP at specific locations fall outside the scope of Atmospheric Chemistry and Physics. In my view the modelling part of the manuscript should be eliminated.*
*The observations are discussed in a more balanced way (with consideration of related work, including appropriate references). The two parts (observations and model) are disjoint and in*

*particular the modelling component is not motivated scientifically and the description of calculations is not sufficiently complete and precise to allow their reproduction as the model setup and out-comes are not discussed in detail.*

We believe that the modeling part of the manuscript completes the observational part, because 1) it provides risk estimates and reference contamination levels related to future nuclear activities in and close to Arctic regions that can be compared to earlier actual releases, and 2) it shows with the Fukushima case how important accurate information on the source term is for the prediction of resulting activity concentrations in the air following an atmospheric release of radioactivity. We have added a note on this justification to the manuscript. What comes to the description of calculations we have given the appropriate literature references concerning the model, the source of the meteorological data and the release parameters. We think that with the information provided, the dispersion calculations can be repeated with any similar computer models.

*The presentation quality of the manuscript (in particular the use of the English language but also the quality of figures and tables) is not of the standard required for publication in ACP.*

Language proofing has been performed for the revised version throughout the text. What becomes to the quality of the figures and tables, as Reviewer 1 did not give specified comments on the tables and figures – whether one or several of these objects do not fill the requirements of the standard level – we will wait if the editorial office gives us instructions for making the possibly required modifications before publishing.

*2 Specific Comments: The abstract provides a concise and complete summary. I propose that the tables listing concentrations are moved to a Supplement, as they are not directly referenced and their inclusion along with the timeline plots in Figures is superfluous.*

In Results & Discussion part, both tables have been referenced several times in case of each nuclide and activity or mass ratio. We see presenting number values in the tables included in the manuscript text as useful because it is more convenient for a reader that the tables are in the same document as text and figures. However, the tables can be published as a separate Supplement part, if the editorial office requires this modification.

*3 Technical Corrections: There are numerous editorial corrections required to reach publication standard, the authors should carefully follow the ACP guide for authors in editing the manuscript before re-submission.*

Any typos or expressions not in line with the ACP manuscript format will be corrected first by the authors in the revised version attached and eventually by the editorial office before publishing the final form.

**Respond to comments of Reviewer 2**

We authors thank Reviewer 2 for giving constructing criticism particularly for our findings of 241Pu from the Fukushima accident to Finland. In the following we will clarify the background how we have interpreted our experimental data into this conclusion, finding Fukushima-derived 241Pu in Finnish Lapland:

The radioanalytical method is briefly reported in the manuscript (page 3, line 85, chapter 2.3 "Measurement of 238,239,240Pu, 241Am, 90Sr, and 240Pu/239Pu in the air filter samples") and the full description is presented separately in another manuscript, which is still under review in MethodsX. We have separated Pu and Am from each other and measured two separate alpha spectra for pure Pu and pure Am. From the activity of daughter nuclide 241Am in the samples we have calculated the activity of mother nuclide 241Pu in the samples back in time for the middle point of the sampling period. So assumptions of Reviewer 2 were partly correct. It is true that time points between sampling, separations and alpha spectrometric measurements of different Pu and Am isotopes, as well as other parameters of the radiochemical analysis need careful attention while determining 241Pu from the ingrowth of 241Am. Apparently some unclear issues have remained in our text and more explanation is needed. Therefore, we added the suggested clarifying text about this issue, measurement of 241Am and 241Pu, to the chapter 2.2 page 3, line 83 and chapter 2.3 at the same page.

The air filter sample set of the year 2011 (collection time was 365 days) was treated (both Pu and Am were separated from the air filter set) in 2014. It means that the short-lived 242Cm (t½ 162.8 d) had decayed 6-7 times its half-life between air sampling and archiving. A mean 242Cm/238Pu activity ratio in soil samples from FDNPP has been determined to be 18 in 11 March 2011 (Povinec et al. Fukushima Accident - Radioactivity Impact on the Environment). According to the same reference, the mean value for 238Pu/239+240Pu activity ratio in the deposition following the Fukushima accident was 2.2. From these values we see that in fresh fallout after the Fukushima accident, the activity of 242Cm was significantly higher than of 238Pu or 239+240Pu. Unfortunately, due to short half-life of 242Cm it was not possible for us to observe 242Cm in the alpha spectrum of 241Am (where also Cm isotopes would be seen due to chemical similarity of Am and Cm) three years after the accident.

The activity ratio 241Pu/239+240Pu is over 100 in deposition from the Fukushima accident (several articles by Jian Zheng et al.). Although the activity ratio 241Pu/239+240Pu could not be determined for our sample of year 2011 due to low amount of 239+240Pu (<3.5 nBq/m3), we think that the presence of Fukushima-derived 241Pu in our air filter sample can't be excluded due to expected (and found) higher amount of 241Pu compared to 239+240Pu. We observed highly increased activity concentration of 241Pu, 602±131 nBq/m3 in 2011, compared with the level of tens nBqs/m3 during years preceding 2011. Therefore, we assume that this increase was caused by the Fukushima accident since there are no other new nuclear events or atmospheric changes as options during time period between Chernobyl and Fukushima accidents.

**(3) Author's changes in manuscript**

Page 2, line 54: a sentence has been added for highlighting the importance of this work and the lack of long time series for transuranium elements at Arctic and Subarctic regions, especially after nuclear accidents in Chernobyl and Fukushima. (Reviewer 1)

Page 2, line 59: An explanation how the modeled nuclear release scenarios are linked to the experimental observation series has been added. (Reviewer 1)

Page 3, line 83: A sentence describing the time when the radiochemical analyses were performed and the time difference between air sampling and radiochemical analyses has been added. (Reviewer 2)

Page 3, line 89: Information about the determination of 241Pu via 241Am and alpha measurements of 241Am has been added, including time information. (Reviewer 2)

Page 10, line 302: A sentence has been added for emphasizing the necessity of ADM as a tool for us, to compare effects of past and possibly forthcoming nuclear events in the atmospheric radionuclide concentrations at Arctic regions. (Reviewer 1)

Page 13, line 400: The significance of the modeled results as a part of our conclusions is explained further in a sentence. (Reviewer 1)

English proofing has been done to the whole text and the resulting small changes are seen with "track changes" throughout the text.

[revised manuscript text omitted]

---

## Author Response (AR2)

Dear Editor,                                                    April 6th 2020

Referring to Your request to check the manuscript once more by a native English speaker, the text has been processed accordingly. Please see the grammatical corrections as "track changes". We hope that the revision is now adequate considering the acceptance of the manuscript.

Best regards,

Susanna Salminen-Paatero & co-workers

[revised manuscript text omitted]